# STRUCTURED PRUNING OF CNNS AT INITIALIZATION

## ABSTRACT

Pruning-at-initialization (PAI) methods prune the individual weights of a convolutional neural network (CNN) before training, thus avoiding expensive fine-tuning or retraining of the pruned model. While PAI shows promising results in reducing model size, the pruned model still requires unstructured sparse matrix computation, making it difficult to achieve a real speedup. In this work, we show both theoretically and empirically that the accuracy of CNN models pruned by a PAI method is independent of the granularity of pruning when *layer-wise density* (i.e., fraction of remaining parameters in each layer) remains the same. We formulate PAI as a convex optimization problem based on an expectation-based proxy for model accuracy, which can instantly produces the optimal allocation of the layer-wise densities under the proxy model. Using our formulation, we further propose a structured and hardware-friendly PAI method, named PreCrop, to prune or reconfigure CNNs in the channel dimension. Our empirical results show that PreCrop achieves a higher accuracy than existing PAI methods on several popular CNN architectures, including ResNet, MobileNetV2, and EfficientNet, on both CIFAR-10 and ImageNet. Notably, PreCrop achieves an accuracy improvement of $1.9\%$ over a state-of-the-art PAI algorithm when pruning MobileNetV2 on ImageNet.

## 1 INTRODUCTION

Convolutional neural networks (CNNs) have achieved state-of-the-art accuracy in a wide range of machine learning (ML) applications. However, the massive computational and memory requirements of CNNs remain a major barrier to more widespread deployment on resource-limited edge and mobile devices. This challenge has motivated a large and active body of research on CNN compression, which attempts to simplify the original model without significantly compromising the accuracy.

Weight pruning LeCun et al. (1990); Han et al. (2015a); Liu et al. (2018); Frankle & Carbin (2018); Han et al. (2015b) has been extensively explored to reduce the computational and memory demands of CNNs. Existing methods create a sparse CNN model by iteratively removing ineffective weights/activations and training the resulting sparse model. Such an iterative pruning approach usually enjoys the least accuracy degradation but at the cost of a more computationally expensive training procedure. Moreover, training-based pruning methods introduce additional hyperparameters, such as the learning rate for fine-tuning and the number of epochs before rewinding Renda et al. (2020), which make the pruning process even more complicated and less reproducible.

To minimize the cost of pruning, a new line of research proposes *pruning-at-initialization (PAI)* Lee et al. (2018); Wang et al. (2020); Tanaka et al. (2020), which identifies and removes unimportant weights in a CNN before training. Similar to training-based pruning, PAI assigns an *importance score* to each individual weight and retains only a subset of them by maximizing the sum of the importance scores of all remaining weights. The compressed model is then trained using the same hyperparameters (e.g., learning rate and the number of epochs) as the baseline model. Thus, the pruning and training of CNNs are cleanly decoupled, greatly reducing the complexity of obtaining a pruned model. Currently, SynFlow Tanaka et al. (2020) is considered the state-of-the-art PAI technique — it eliminates the need for data during pruning as required in prior arts Lee et al. (2018); Wang et al. (2020) and achieves a higher accuracy with the same compression ratio.

However, existing PAI methods mostly focus on fine-grained weight pruning, which removes individual weights from the CNN model without preserving any structures. As a result, both inference and training of the pruned model require sparse matrix computation, which is challenging to accelerate

on commercially-available ML hardware that is optimized for dense computation (e.g., GPUs and TPUs Jouppi et al. (2017)). According to a recent study Gale et al. (2020), even with the NVIDIA cuSPARSE library, one can only achieve a meaningful speedup for sparse matrix multiplications on GPUs when the sparsity is over 98%. In practice, it is difficult to compress modern CNNs by more than $50\times$ without a drastic degradation in accuracy Blalock et al. (2020). Therefore, structural pruning patterns (e.g., pruning weights for the entire output channel) are preferred to enable practical memory and computational saving by avoiding irregular sparse storage and computation.

In this work, we propose novel structured PAI techniques and demonstrate that they can achieve the same level of accuracy as the unstructured methods. We first introduce **synaptic expectation (SynExp)**, a new proxy metric for accuracy, which is defined to be the expected sum of the importance scores of all the individual weights in the network. SynExp is invariant to weight shuffling and reinitialization, thus addressing some of the deficiencies of the fine-grained PAI approaches found in recent studies Su et al. (2020); Frankle et al. (2020). We also show that SynExp does not vary as long the layer-wise density remains the same, irrespective of the granularity of pruning. Based on this key observation, we formulate an optimization problem that maximizes SynExp to determine the layer-wise pruning ratios, subject to model size and/or FLOPs constraints. We then propose **PreCrop**, a structured PAI that prunes CNN models at the channel level in a way to achieve the target layer-wise density determined by the SynExp optimization. PreCrop can effectively reduce the model size and computational cost without loss of accuracy compared to existing fine-grained PAI methods. Besides channel-level pruning, we further propose **PreConfig**, which can *reconfigure* the width dimension of a CNN to achieve a better accuracy-complexity trade-off with almost zero computational cost. Our empirical results show that the model after PreConfig can achieve higher accuracy with fewer parameters and FLOPs than the baseline for a variety of modern CNN architectures.

We summarize our contributions as follows:

- We propose to use the SynExp as a proxy for accuracy and formulate PAI as an optimization problem that maximizes SynExp under model size and/or FLOPs constraints. We show that the accuracy of the CNN model pruned by solving the constrained optimization is independent of the pruning granularity.

- We introduce PreCrop, a channel-level structured pruning technique that builds on the proposed SynExp optimization.

- We show that PreConfig can be used to optimize the width of each layer in the network with almost zero computational cost (e.g., within one second on CPU).

## 2 RELATED WORK

**Training-Based Pruning** uses various heuristic criteria to prune unimportant weights. They typically employ an iterative training-prune-retrain process where the pruning stage is intertwined with the training stage, which may increase the overall training cost by several folds.

Existing training-based pruning methods can be either unstructured Han et al. (2015a); LeCun et al. (1990) or structured He et al. (2017); Luo et al. (2017), depending on the granularity and regularity of the pruning scheme. Training-based unstructured pruning usually achieves better accuracy given the same model size budget, while structured pruning can achieve a more practical speedup and compression without special support from custom hardware.

**(Unstructured) Pruning-at-Initialization (PAI)** Lee et al. (2018); Wang et al. (2020); Tanaka et al. (2020) methods provide a promising approach to mitigating the high cost of training-based pruning. They can identify and prune unimportant weights after initialization and before training starts. Related to these efforts, authors of Frankle et al. (2020) and Su et al. (2020) independently find that for the existing PAI methods, randomly shuffling the weights within a layer or reinitializing the weights does not cause any accuracy degradation. Liu et al. (2022) provides evidence that random pruning can achieve comparable accuracy to other pruning methods, given a suitable layerwise pruning ratio.

**Additional Related Work** This work also has connections to other domains, such as Neural Architecture Search (NAS) and other pruning techniques. For further discussion on these related topics, please refer to Appendix A.

Figure 1: Illustration of PreCrop — The entire pruning process takes less than 1 second on a CPU. Further details on [(1)]SynExp Optimization and [(2)]Channel Cropping will be introduced in Section 3 and 4, respectively. The standard training procedure is employed once the model is pruned by PreCrop.

## 3 PRUNING-AT-INITIALIZATION VIA SYNEXP OPTIMIZATION

In this section, we first review the preliminaries and deficiencies of existing PAI methods. To overcome the limitations, we introduce a new proxy for the accuracy of the PAI compressed model. We then propose a new formulation of PAI that maximizes the proxy metric using convex optimization.

### 3.1 PAI BACKGROUND

**Preliminaries.** PAI aims to prune a neural network after initialization but before training to avoid the time-consuming *training-pruning-retraining* process. Prior to training, PAI typically uses the magnitude of gradients (with respect to weights) to estimate the importance of individual weights. This requires both forward and backward propagation passes. PAI prunes the weights ($W$) with smaller importance scores by setting the corresponding entries in the binary weight mask ($M$) to zero. More concretely, to remove weights $W$, $M$ is applied to $W$ in an element-wise manner as $W \odot M$, where $\odot$ denotes the Hadamard product.

Popular PAI approaches, such as SNIP Lee et al. (2018), GraSP Wang et al. (2020), and SynFlow Tanaka et al. (2020), employ different methods to estimate the importance of individual weights. SNIP and GraSP, as Single-shot PAI algorithms, prune the model to the desired sparsity in a single pass. SynFlow, which represents the state-of-the-art PAI algorithm, repetitively prunes a small fraction of weights and re-evaluating the importance scores until the desired pruning rate reached. Through the iterative process, the importance of each weight can be estimated more accurately.

Specifically, the importance score for a fully connected network used in SynFlow is defined as:

$$\mathcal{S}(W_{ij}^l) = \left[ \mathbb{1}^T \prod_{k=l+1}^{N} \left| W^k \odot M^k \right| \right]_i \left| W_{ij}^l M_{ij}^l \right| \left[ \prod_{k=1}^{l-1} \left| W^k \odot M^k \right| \mathbb{1} \right]_j , \tag{1}$$

where $N$ is the number of layers, $W^l$ and $M^l$ are the weight and weight mask of the $l$-th layer, $\mathcal{S}(W_{ij}^l)$ is the SynFlow score for a single weight $W_{ij}^l$, $| \cdot |$ is element-wise absolute operation, and $\mathbb{1}$ is an all-one vector. Here no training data or labels are required to compute the importance score, thus making SynFlow a data-agnostic algorithm.

**Deficiencies.** Similar to training-based fine-grained pruning Han et al. (2015a); LeCun et al. (1990), existing PAI methods also use the *sum* of importance scores of the remaining weights as a proxy for model accuracy. Specifically, PAI obtains a binary weight mask (i.e., the pruning decisions) by maximizing the following objective:

$$\text{maximize} \sum_{l=1}^{N} \mathcal{S}^l \cdot M^l \quad \text{over } M^l, \quad \text{subject to} \sum_{l=1}^{N} \|M^l\|_0 \leq B_{\text{params}} , \tag{2}$$

where $S^l$ is the importance score matrix for the $l$-th layer, $\| \cdot \|_0$ is the number of nonzero entries in a matrix, and $B_{\text{params}}$ is the target size of the compressed model.

Given the setup of this optimization, it is natural that a subset of the individual weights will be deemed more important than others. Moreover, existing methods for computing the importance scores all depend on the values of the weights, thus any updates to the weights (such as reinitialization) will easily result in a change to the accuracy metric (i.e., the sum of the individual importance scores).

Nevertheless, recent studies in Su et al. (2020); Frankle et al. (2020); Liu et al. (2022) has demonstrated the significant impact of layerwise density on the accuracy of PAI. This finding suggests that the aforementioned metric is not a good proxy for indicating the accuracy of the pruned model.

## 3.2 SYNEXP INVARIANCE THEOREM

In this section, we propose a new proxy metric called SynExp to address the deficiencies of the existing PAI approaches. We argue that a good accuracy proxy should achieve the following properties:

1. The pruning decision (i.e., weight mask $M$) can be made *before* the model is initialized.

2. Maximization of the proxy should output *layer-wise density* $p_l$ as the result, as opposed to pruning decisions for individual weights.

For the ease of later discussion, we formalize the weight matrix $W$ and weight mask matrix $M$ as two random variables, given a fixed density $p_l$ for each layer, for random pruning before initialization. If $W^l$ contains $\alpha_l$ parameters, $A^l = \{M, M_i^l \in \{0, 1\} \; \forall 1 \le i \le \alpha_l, \sum_i M_i^l = p_l \times \alpha_l\}$ is the set of all possible $M^l$ with the same shape as the $W^l$ that satisfies the layer-wise density ($p_l$) constraint. Then, the random weight mask $M^l$ for layer $l$ is sampled uniformly from $A^l$. Also, each individual weight $W_i^l$ in layer $l$ is independently sampled from a given distribution $D^l$.

The observations in Section 3.1 indicate that different values of these two random variables $M$ and $W$ result in similar final accuracy of the pruned model. However, different values do change the proxy value for the model accuracy in existing PAI methods. For example, the SynFlow score in Equation 2 may change under different instantiations of $M$ and $W$. Therefore, we propose a new proxy that is invariant to the instantiation of $M$ and $W$ for the model accuracy in the context of PAI — the *expectation* of the sum of the importance scores of all unpruned (i.e., remaining) weights. The proposed proxy can be formulated as follows:

$$\text{maximize} \; \mathop{\mathbb{E}}_{M,W}[\mathcal{S}] = \mathop{\mathbb{E}}_{M,W}\left[\sum_{l=1}^{N} S^l \cdot M^l\right] \; \text{over } p_l \quad \text{subject to} \; \sum_{l=1}^{N} \alpha_l \cdot p_l \le B_{\text{params}} \;, \quad (3)$$

where $\mathbb{E}_{M,W}[\mathcal{S}]$ stands for the expectation of the importance score $\mathcal{S}$ over $W$ and $M$. In this new formulation, $p_l$ is optimized to maximize the proposed proxy for model accuracy. Since the expectation is computed over the $W$ and $M$, the instantiations of these two random variables do not affect the expectation.

Moreover, we prove that, in comparison with the SynFlow score, the utilization of SynExp leads to a smaller error associated with predicting accuracy, The complete proof can be found in Appendix C.

To evaluate the expectation before weight initialization, we adopt the importance metric proposed by SynFlow, i.e., plugging $\mathcal{S}$ in Equation 1 into Equation 3 As a result, we can compute the expectation analytically without forward or backward propagations. This new expectation-based proxy is referred to as *SynExp*, i.e., synaptic expectation.

We show SynExp is invariant to the granularity of pruning PAI in the SynExp Invariance Theorem, which is stated as follows.

**Theorem 3.1. SynExp Invariance Theorem.** Given a specific CNN architecture, the SynExp ($\mathbb{E}_{[M,W]}[\mathcal{S}_{\text{SF}}]$) of any randomly compressed model with the same layer-wise density $p_l$ is a constant, independent of the pruning granularity. The constant SynExp equals:

$$\mathop{\mathbb{E}}_{M,W}[\mathcal{S}_{\text{SF}}] = NC_{N+1}\prod_{l=1}^{N}(p_l C_l \cdot \mathbb{E}_{x \sim \mathcal{D}}[|x|]) \;, \quad (4)$$

where $N$ is the number of layers in the network, $\mathbb{E}_{x \sim \mathcal{D}}[|x|]$ is the expectation of magnitude of distribution $\mathcal{D}$, $C_l$ is the input channel size of layer $l$ and is also the output channel size of $l-1$, and $p_l = \frac{1}{\alpha_l}\|M_l\|_0$ is the layer-wise density.

In Equation 4, $N$ and $C_l$ are all hyperparameters of the CNN architecture and can be considered constants. $\mathbb{E}_{|\mathcal{D}^l|}$ is also a constant under a particular distribution $\mathcal{D}^l$. The layer-wise density $p_l$ is

the only variable in the equation. Thus, SynExp satisfies both of the aforementioned properties: 1) pruning is done prior to the weight initialization; 2) the layer-wise density can be directly optimized. Furthermore, Theorem 1 shows that the granularity of pruning has no impact on the proposed SynExp metric. In other words, the CNN model compressed using either unstructured or structured pruning method is expected to have a similar accuracy.

The detailed proof of SynExp Invariance Theorem can be found in Appendix B. We also empirically verify it by randomly pruning each layer of a CNN at three different granular levels but with the same layer-wise density. Specifically, we perform random pruning at (1) weight-, (2) filter-, and (3) channel-level to achieve the desired layer-wise pruning ratios obtained from solving Equation 3. For weight and filter pruning, randomly pruning each layer to match the layer-wise density $p_l$ occasionally detaches some weights from the network, especially when the density is low. The detached weights do not contribute to the prediction but are counted as remaining parameters. Thus, we remove the detached weights for a fair comparison following the same approach described in Vysogorets & Kempe (2021). For channel pruning, it is not trivial to achieve the target layer-wise density

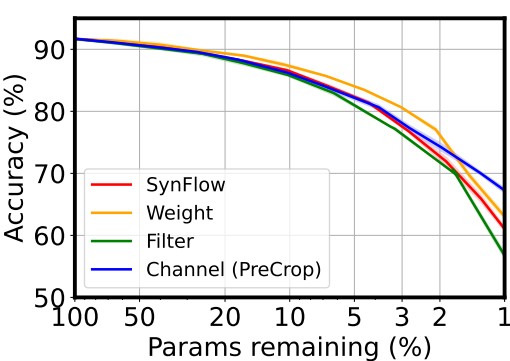

Figure 2: Comparison of the performance using different pruning granularities on ResNet20 using CIFAR-10.

while satisfying the constraint that the number of output channels of the previous layer must equal the number of input channels of the next layer. Therefore, we employ PreCrop proposed in Section 4.2.

As shown in Figure 2, random pruning with different granularities can obtain a similar accuracy compared to SynFlow, as long as the layer-wise density remains the same. The empirical results are consistent with SynExp Invariance Theorem and also demonstrate the efficacy of the proposed SynExp metric. In Appendix E, we enhance the validation of our theorem with the same experiment using layerwise density obtained by different methods (i.e., SNIP, GraSP, and ERK Evci et al. (2020))

### 3.3 OPTIMIZING SYNEXP

As discussed in Section 3.2, the layer-wise density matters for our proposed SynExp approach. Here, we show how to obtain the layer-wise density in Equation 3 that maximizes SynExp under model size and/or FLOPs constraints.

#### 3.3.1 OPTIMIZING SYNEXP WITH PARAMETER COUNT CONSTRAINT

Given that the goal of PAI is to reduce the size of the model, we need to add a constraint on the total number of parameters $B_{\text{params}}$ (i.e., parameter count constraint), where $B_{\text{params}}$ is typically greater than zero and less than the number of parameters in the original network. Since layer-wise density $p_l$ is the only variable in Equation 3, we can simplify the equation by removing other constant terms, as follows:

$$\text{maximize} \sum_{l=1}^{N} \log p_l \quad \text{over } p_l, \quad \text{subject to} \quad \begin{aligned} \sum_{l=1}^{N} \alpha_l \cdot p_l &\leq B_{\text{params}} \,, \\ 0 < p_l &\leq 1, \forall 1 \leq l \leq N \,, \end{aligned} \tag{5}$$

where $\alpha_l$ is the number of parameters in layer $l$.

Equation 5 is a convex optimization problem that can be solved analytically[1]. We compare the layer-wise density derived from solving Equation 5 with the density obtained using SynFlow. It is also worth noting that the proposed method can find the optimal layer-wise density even before the network is initialized.

### 3.3.2 Optimizing SynExp with Parameter Count and FLOPs Constraints

As discussed in Section 3.3.1, we can formulate PAI as a convex optimization problem with a constraint on the model size. However, the number of parameters does not necessarily reflect the performance (e.g., throughput) of the CNN model. In many cases, CNN models are compute-bound on commodity hardware Jouppi et al. (2017); Harish & Narayanan (2007). Therefore, we also introduce a FLOPs constraint in our formulation.

With the existing PAI algorithms, it is not straightforward to directly constrain the optimization using a bound on the FLOP count. The savings in FLOPs are instead the byproduct of the weight pruning as specified in Equation 2. In contrast, it is much easier to account for FLOPs in our formulation, which aims to determine the density of each layer as opposed to the inclusion of individual weights from different layers. Thus for each layer, we can easily derive the required FLOP count based on the density ($p_l$). After incorporating the constraint on FLOPs ($B_{\text{FLOPs}}$), the convex optimization problem becomes:

$$\text{maximize} \sum_{l=1}^{N} \log p_l \quad \text{over } p_l, \quad \text{subject to} \quad \sum_{l=1}^{N} \alpha_l \cdot p_l \leq B_{\text{params}}, \quad \sum_{l=1}^{N} \beta_l \cdot p_l \leq B_{\text{FLOPs}}, \quad (6)$$
$$0 < p_l \leq 1, \forall 1 \leq l \leq N,$$

where $\beta_l$ is the number of FLOPs in the $l^{th}$ layer.

Since the additional FLOPs constraint is linear, the optimization problem in Equation 6 remains convex and has an analytical solution[1]. By solving SynExp optimization with a fixed $B_{\text{params}}$ but different $B_{\text{FLOPs}}$, we can obtain the layer-wise density for various models that have the same number of parameters but different FLOPs. Then, we perform random weight pruning on the CNN model to achieve the desired layer-wise density. We compare the proposed SynExp optimization (denoted as Ours) with other popular PAI methods. As depicted in Figure 3, given a fixed model size ($1.5 \times 10^4$ in the figure), our method can be used to generate a Pareto Frontier that spans the spectrum of FLOPs, while other methods can only have a fixed FLOPs. Our method dominates all other methods in terms of both accuracy and FLOPs reduction.

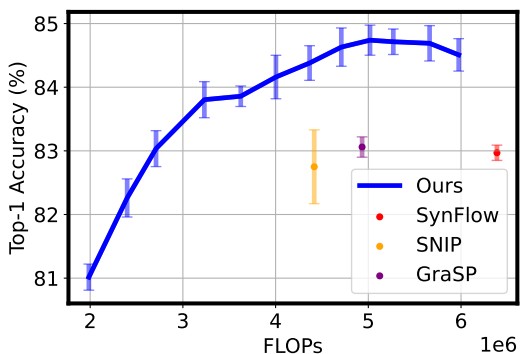

Figure 3: Comparison of our method with other PAI methods — we repeat the experiment using ResNet-20 on CIFAR-10 five times and report the mean and variance (error bar) of the accuracy. All the models in the figures have $1.5 \times 10^4$ parameters.

## 4 Structured Pruning-at-Initialization

The SynExp Invariance Theorem shows that the pruning granularity of PAI methods should not affect the accuracy of the pruned model. Channel pruning, which prunes the weights of the CNN at the output channel granularity, is considered the most coarse-grained and hardware-friendly pruning technique, Therefore, applying the proposed PAI method for channel pruning can avoid both complicated retraining/re-tuning procedures and irregular computations. In this section, we propose a structured PAI method for channel pruning, named PreCrop, to prune CNNs in the channel dimension. In addition, we propose a variety of PreCrop with relaxed density constraints to reconfigure the width of each layer in the CNN model, which is called PreConfig.

### 4.1 PreCrop

Applying the proposed PAI method to channel pruning requires a two-step procedure. First, the layer-wise density $p_l$ is obtained by solving the optimization problem shown in Equation 5 or 6. Second, we need to decide how many output channels of each layer should be pruned to satisfy

---

[1]We include analytical solutions for Equation 5 and Equation 6 in Appendix D for completeness.

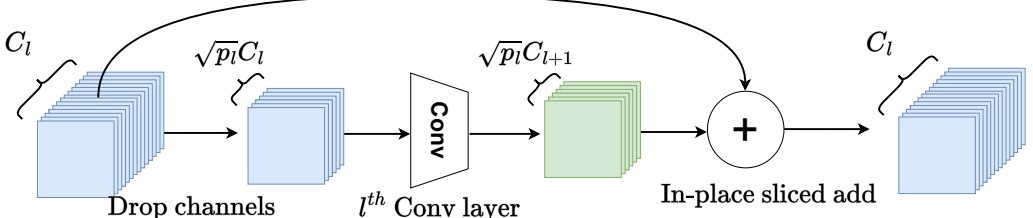

Figure 4: Illustration of PreCrop for layers with residual connections[2] — $C_l$ and $C_{l+1}$ represent the number of input channels of layer $l$ and $l + 1$. $p_l$ represents the density of layer $l$.

the layer-wise density. However, it is not straightforward to compress each layer to match a given layer-wise density due to the additional constraint that the number of output channels of the current layer must match the number of input channels of the next layer.

We introduce PreCrop, which compresses each layer to meet the desired layer-wise density. Let $C_l$ and $C_{l+1}$ be the number of input channels of layer $l$ and $l + 1$, respectively. $C_{l+1}$ is also the number of output channels of layer $l$. For layers with no residual connections, the number of output channels of layer $l$ is reduced to $\lfloor \sqrt{p_l} \cdot C_{l+1} \rfloor$. The number of input channels of layer $l + 1$ needs to match the number of output channels of layer $l$, which is also reduced to $\lfloor \sqrt{p_l} \cdot C_{l+1} \rfloor$. Therefore, the actual density of layer $l$ after PreCrop is $\sqrt{p_{l-1} \cdot p_l}$ instead of $p_l$. We empirically find that $\sqrt{p_{l-1} \cdot p_l}$ is close enough to $p_l$ because the neighboring layers have similar layer-wise densities. Alternatively, one can obtain the exact layer-wise density $p$ by only reducing the number of input or output channels of a layer. However, this approach leads to a significant drop in accuracy, because the number of the input and output channels can change dramatically (e.g., $p_l C_l \ll C_{l+1}$ or $C_l \gg p_l C_{l+1}$). This causes the shape of the feature map to change dramatically in adjacent layers, resulting in information loss.

For layers with residual connections, we use in-place sliced addition to address the constraint on the number of channels of adjacent layers, as shown in Figure 4. We reduce the number of input and output channels of layer $l$ from $C_l$ and $C_{l+1}$ to $\sqrt{p_l}C_l$ and $\sqrt{p_l}C_{l+1}$. Thus the density of each layer matches the given layer-wise density. Subsequently, we use in-place sliced addition[2] to address the mismatch between the channel numbers of the original input to layer $l$ and the output of layer $l$. PreCrop eliminates the requirement for sparse computation in existing PAI methods and thus can be used to accelerate both training and inference of the pruned models.

## 4.2 PreConfig: PreCrop with Relaxed Density Constraint

PreCrop uses the layer-wise density obtained from solving the convex optimization problem, which is always less than 1 following the common setting for pruning (i.e., $p_l \leq 1$). However, this constraint is not necessary for our method since we can increase the number of channels (i.e., expand the width of the layer) before initialization. By solving the problem in Equation 6 without the constraint $p_l \leq 1$, we can expand the layers with a density greater than 1 ($p_l > 1$) and prune the layers with a density less than 1 ($p_l < 1$). We call this variant of PreCrop as PreConfig (PreCrop-Reconfigure). If we set $B_{\text{params}}$ and $B_{\text{FLOPs}}$ to be the same as the original network, we can essentially reconfigure the width of each layer of a given network architecture under certain constraints on model size and FLOPs.

The width of each layer in a CNN is usually designed manually, which often relies on extensive experience and intuition. Using PreConfig, we can automatically determine the width of each layer in the network to achieve a better cost-accuracy trade-off. PreConfig can also be used as (a part of) an ultra-fast NAS. Compared to conventional NAS, which typically searches on the width, depth, resolution, and choice of building blocks, PreConfig only changes the width. Nonetheless, PreConfig only requires a minimum amount of time and computation compared to NAS methods; it only needs to solve a relatively small convex optimization problem, which can finish within a second on a CPU.

## 5 Evaluation

In this section, we empirically evaluate PreCrop and PreConfig.

---

[2]In-place sliced addition between tensor `A` with `c1` channels and tensor `B` with `c2` channels (`c1 < c2`) can be efficiently implemented by `A[:c1] += B`

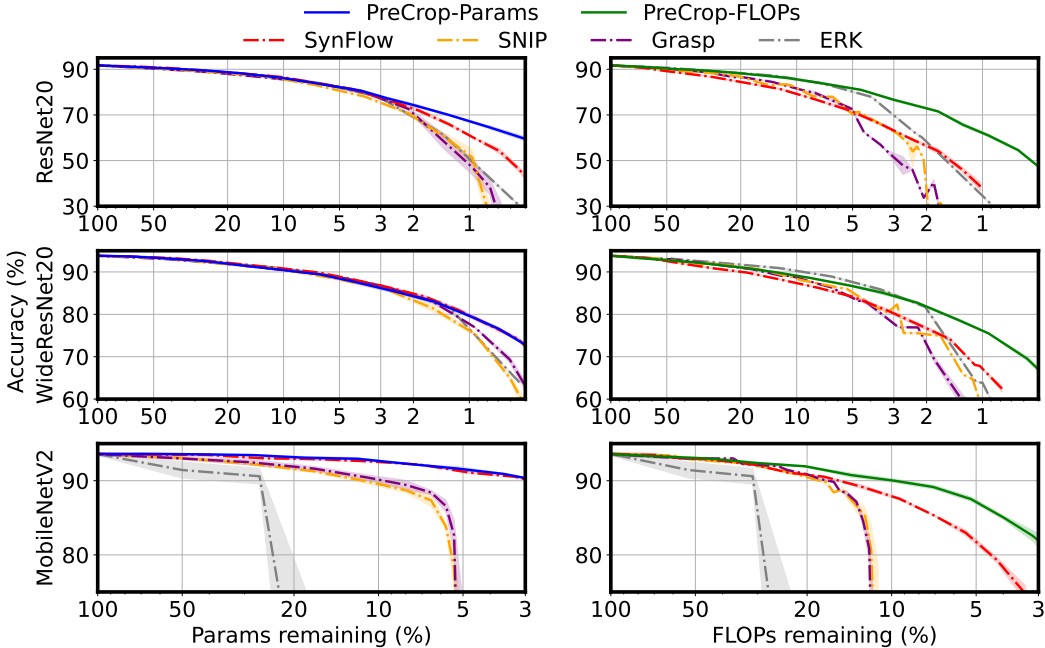

Figure 5: Comparison of PreCrop-Params and PreCrop-FLOPs with baselines — we repeat the experiment using ResNet20 (top), WideResNet20 (middle), and MobileNetV2 (bottom) on CIFAR-10 three times and report the mean and variance (error bar) of the accuracy.

We limit our comparison to unstructured PAI methods since their pruning costs are similar to our approach. We first demonstrate the effectiveness of PreCrop by comparing it with baselines on CIFAR-10. Baseline unstructured PAI methods include SynFlow, SNIP, Grasp, and ERK. We then use PreConfig to tune the width of each layer and compare the accuracy of the model after PreConfig with the original model. We perform experiments using various modern CNN models, including ResNet He et al. (2016), MobileNetV2 Sandler et al. (2018), and EfficientNet Tan & Le (2019), on both CIFAR-10 and ImageNet. We set all hyperparameters used to train the models pruned by different PAI algorithms to be the same. Detailed experimental settings can be found in Appendix G.

## 5.1 EVALUATION OF PRECROP

For CIFAR-10, we compare the accuracy two variants of PreCrop: PreCrop-Params (blue line) and PreCrop-FLOPs (green line) with baseline unstructured PAI methods (dashed line). PreCrop-Params adds the parameter count constraint whereas PreCrop-FLOPs imposes the FLOPs constraint into the convex optimization problem. As shown in Figure 5, PreCrop-Params achieves similar or even better accuracy as the best baseline, SynFlow, under a wide range of different model size constraints, thus validating that PreCrop-Params can be as effective as the fine-grained PAI method. Considering the benefits of structured pruning, PreCrop-Params should be favored over existing PAI methods. It further demonstrates that PreCrop-FLOPs outperforms all baselines by a large margin, especially when FLOPs sparsity is high.

Table 1 summarizes the comparison between PreCrop and SynFlow on ImageNet. For ResNet-50, PreCrop achieves 0.2% lower accuracy compared to SynFlow with similar model size and FLOPs. For both MobileNetV2 and EfficientNetB0, PreCrop achieves 1.9% and 1.0% accuracy improvements compared to SynFlow with strictly fewer FLOPs and parameters, respectively. The experimental results on ImageNet further support SynExp Invariance Theorem that coarse-grained structured pruning (e.g., PreCrop) can perform as well as unstructured pruning. In conclusion, PreCrop achieves a favorable accuracy and model size/FLOPs tradeoff compared to the state-of-the-art PAI algorithm.

## 5.2 EVALUATION OF PRECONFIG

Table 2 compares the accuracy of the reconfigured model with the original model under similar model size and FLOPs constraints. For ResNet-50, with similar accuracy, we reduce the parameter count

Table 1: **Comparison of PreCrop with SynFlow on ImageNet** — The dagger($^\dagger$) implies that the numbers are theoretical without considering the overhead of sparse matrices in storing and computing.

| NETWORK | METHODS | FLOPs (G) | PARAMS (M) | ACCURACY (%) |
|---|---|---|---|---|
| RESNET50 | BASELINE | 3.85 | 25.6 | 76.3 |
| | SYNFLOW | 2.82$^\dagger$ | 12.81$^\dagger$ | 73.9 (-2.4) |
| | PRECROP | **2.80** | **12.76** | **75.2 (-1.1)** |
| MOBILENETV2 | BASELINE | 0.33 | 3.51 | 72.4 |
| | SYNFLOW | **0.26$^\dagger$** | **2.44$^\dagger$** | 70.6 (-1.8) |
| | PRECROP | **0.26** | **2.44** | **71.2 (-1.2)** |
| | SYNFLOW | **0.21$^\dagger$** | **1.91$^\dagger$** | 67.9 (-4.5) |
| | PRECROP | **0.21** | **1.91** | **69.8 (-2.6)** |
| EFFICIENTNETB0 | BASELINE | 0.40 | 5.29 | 75.7 |
| | SYNFLOW | **0.30$^\dagger$** | 3.72$^\dagger$ | 73.4 (-2.5) |
| | PRECROP | **0.30** | **3.67** | **74.2 (-1.5)** |

Table 2: **PreConfig on ImageNet.**

| NETWORK | METHODS | FLOPs (G) | PARAMS (M) | ACCURACY (%) |
|---|---|---|---|---|
| RESNET50 | BASELINE | 3.85 | 25.56 | **76.3** |
| | PRECONFIG | **3.81 (98.9%)** | **22.31 (87.3%)** | 76.1 (-0.2) |
| MOBILENETV2 | BASELINE | 0.33 | 3.51 | **72.4** |
| | PRECONFIG | **0.32 (97.0%)** | **2.89 (82.3%)** | **72.4** |
| EFFICIENTNETB0 | BASELINE | **0.40** | 5.29 | 75.7 |
| | PRECONFIG | **0.40** | **5.03 (95.1%)** | **75.8 (+0.1)** |

by 25%. For MobileNetV2, we achieve $0.3\%$ higher accuracy than the baseline with $20\%$ fewer parameters and $3\%$ fewer FLOPs. For EfficientNet identified by NAS method, we can also achieve $0.1\%$ higher accuracy than the baseline with $5\%$ fewer parameters and the same FLOPs.

## 5.3 SPEEDUP IN WALL-CLOCK TIME

PreCrop also exhibits hardware-friendly characteristics. Our experiments demonstrate that applying PreCrop to MobileNetV2 and EfficientNet models results in wall-clock speedup for both inference and training. Detailed experimental settings can be found in Appendix G.4.

Specifically, the inference latency is accelerated by $1.35\times$ and $1.11\times$ for MobileNetV2 and EfficientNetB0, respectively, as shown in Table 3. The training latency also has a speedup of $1.27\times$ and $1.12\times$ for the respective models. Note that

Table 3: **Wall clock speedup comparison.** Measured with an NVIDIA V100 GPU.

| | Inf. Lat. (ms) | Train Lat. (ms) |
|---|---|---|
| | MobileNetV2 | |
| Baseline | 42.5 | 141 |
| PreCrop | **31.5 (1.35$\times$)** | **111 (1.27$\times$)** |
| | EfficientNetB0 | |
| Baseline | 58 | 235 |
| PreCrop | **52 (1.11$\times$)** | **212 (1.12$\times$)** |

PreCrop achieves a FLOPs reduction of $1.57\times$ for MobileNetV2 and $1.33\times$ for EfficientNetB0. The observed gap between wall-clock speedup and FLOPs reduction can be attributed to the sub-optimal parallelism and optimization for smaller layers, as noted in affiliates (2021).

## 6 CONCLUSION

In this work, we theoretically and empirically show that the accuracy of the CNN models pruned using PAI methods does not depend on pruning granularity. We formulate PAI as a simple convex SynExp optimization. Based on SynExp optimization, we further propose PreCrop and PreConfig to prune and reconfigure CNNs in the channel dimension. Our experimental results demonstrate that PreCrop can outperform existing fine-grained PAI methods on various networks and datasets while achieving wall-clock time speedups for both training and inference stages.

## REPRODUCIBILITY STATEMENT

The proof of SynExp Invariance Theorem is stated in the appendix with explanations. We provide the source code for the key experiments in the paper. We thoroughly checked the implementation and also verified empirically that the results in this paper are reproducible. The source code will be made available through GitHub.

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

## A  EXTENDED RELATED WORK

**Neural Architecture Search (NAS)** Zoph & Le (2016); Wan et al. (2020) automatically explores a large space of candidate models to achieve a better accuracy-efficiency trade-off. The typical bases of the NAS search space include the width, depth, resolution, and choice of building blocks. However, existing approaches can only search in a small subset of the possible configurations due to the cost. For example, the search space of the channel width usually only contains a limited set of integer values. The cost for NAS is also orders of magnitude higher than training a single model.

In this work, the structured pruning can be done within 1 second before the training and inference stages, resulting in improved performance in both stages. Some NAS algorithms Abdelfattah et al. (2021); Zhou et al. (2020); Zhang & Jia (2022); Mellor et al. (2021); Zhou et al. (2022) utilizes a less computationally expensive proxy rather than training the whole network, but still require an expensive reinforcement learning Abdelfattah et al. (2021) or evolutionary algorithm Real et al. (2019) to predict a good network. The overall cost of NAS remains comparable to that of the training process.

**More PAI Methods** Another iterative PAI approach proposed in Verdenius et al. (2020) which shares a core concept similar to that in SynFlow Tanaka et al. (2020). Additionally, De Jorge et al. (2020) put forward a structured PAI technique. However, unlike this work, they utilized distinct formulations for structured and unstructured pruning and did not establish links between different pruning granularities.

**Other Pruning Techniques** In Section 2, we discussed two kinds of pruning methods: training-based pruning and pruning-at-initialization (PAI). However, there are some other pruning techniques Rachwan et al. (2022); Liu et al. (2021.); YVINEC et al. (2021) that span the continuum between these two pruning approaches. These methodologies seek to reduce computational costs by implementing pruning during the training phase.

For instance, Rachwan et al. (2022) proposed EarlyCroP, a technique that extracts sparse models at the early stages of training. Similarly, Liu et al. (2021.) introduced Gradual Pruning with Zero-cost Neuroregeneration (GraNet), a during-training pruning method, which improves pruning plasticity and advances the state of the art by boosting sparse-to-sparse training performance. YVINEC et al. (2021), on the other hand, presented RED, a data-free approach that tackles structured pruning by merging redundant neurons and employing a novel uneven depthwise separation technique for further pruning convolutional layers.

Despite their novelty, these methods still result in a higher computational burden during the early stages of training, making them less comparable to the PAI methods.

## B  PROOF OF SYNEXP INVARIANCE THEOREM

**Theorem B.1.** *Given a specific CNN architecture, the SynExp ($\mathbb{E}_{[M,W]}[\mathcal{S}_{SF}]$) of any randomly compressed model with the same layer-wise density $p_l$ is a constant, independent of the pruning granularity. The constant SynExp equals:*

$$\mathop{\mathbb{E}}_{M,W}[\mathcal{S}_{SF}] = NC_{N+1} \prod_{l=1}^{N} (p_l C_l \cdot \mathbb{E}_{x \sim \mathcal{D}}[|x|]) , \tag{7}$$

*where $N$ is the number of layers in the network, $\mathbb{E}_{x \sim \mathcal{D}}[|x|]$ is the expectation of magnitude of distribution $\mathcal{D}$, $C_l$ is the input channel size of layer $l$ and is also the output channel size of $l-1$, and $p_l = \frac{1}{\alpha_l} \|C_l\|_0$ is the layer-wise density.*

*Proof.* Assuming the network has $N$ layers, weight matrix $W^l \in \mathbb{R}^{C_l \times C_{l+1}}$, mask matrix $M^l \in \{0, 1\}^{C_l \times C_{l+1}}$. $C_l$ and $C_{l+1}$ are the input and output channel size of layer $l$. As the output channel size of any layer $l$ equals to the input channel size of the next layer $l+1$, we have $C_{l+1} = C_{l+1}, \forall l < N$.

We first prove the Theorem B.1 on fully-connected network, and we can extend it to CNNs easily. From Equation 1, in a fully-connected network, the Synaptic Flow score for any parameter $W_{ij}^l$ with

mask $M_{ij}^l$ in layer $l$ equals to:

$$\mathcal{S}_{\text{SF}}(W_{(i,j)}^l) = \left[ \mathbb{1}^T \prod_{k=l+1}^N |W^k \odot M^k| \right]_i \left| W_{(i,j)}^l M_{(i,j)}^l \right| \left[ \prod_{k=1}^{l-1} |W^k \odot M^k| \mathbb{1} \right]_j \quad (8)$$

We compute the SynExp of the layer $l$ ($\mathbb{E}_{[M,W]}(\mathcal{S}_{\text{SF}})^{[l]}$), then the SynExp of the network is simply the sum of SynExp of all layers:

$$\mathbb{E}_{[M,W]}(\mathcal{S}_{\text{SF}}) = \sum_{l=1}^N \mathbb{E}_{[M,W]}(\mathcal{S}_{\text{SF}})^{[l]} \quad (9)$$

We define the expectation value for input channel $i$, output channel $j$, and the whole layer in layer $l$ as $E_{(i,*)}^l$, $E_{(*,j)}^l$, and $E_{(*,*)}^l$:

$$\mathbb{E}_{(i,*)}^l = \frac{1}{C_{l+1}} \sum_x |W_{(i,x)}^l M_{(i,x)}^l| \quad (10)$$

$$\mathbb{E}_{(*,j)}^l = \frac{1}{C_l} \sum_x |W_{(x,j)}^l M_{(x,j)}^l| \quad (11)$$

$$\mathbb{E}_{(i,j)}^l = \mathbb{E}_{(*,*)}^l = \frac{1}{C_l C_{l+1}} \sum_{i,j} |W_{(i,j)}^l M_{(i,j)}^l| = \frac{1}{\alpha_l} \sum_{i,j} |W_{(i,j)}^l M_{(i,j)}^l| = p_l \mathbb{E}_{|\mathcal{D}^l|} \quad (12)$$

Here we use $\mathbb{E}_{|\mathcal{D}^l|}$ to denote $\mathbb{E}_{x \sim \mathcal{D}}[|x|]$.

As the weight in layer $l$ is sampled from distribution $\mathcal{D}$, and the mask matrices are also randomly sampled, we have

$$\mathbb{E}_{(*,*)}^{[k]} = \mathbb{E}_{(i,*)}^k = \mathbb{E}_{(*,j)}^k = p_l \mathbb{E}_{|\mathcal{D}^l|} \quad (13)$$

With $E_{(i,*)}^k$, $E_{(*,j)}^k$, and $E_{(*,*)}^l$, we can rewrite Equation 8 to:

$$\mathbb{E}[\mathcal{S}_{\text{SF}}(W_{(i,j)}^l)] = \left( \prod_{k=l+2}^N C_{k+1} \mathbb{E}_{(*,*)}^k \right) \cdot C_{l+2} \mathbb{E}_{(i,*)}^{l+1} \cdot \mathbb{E}_{(i,j)}^l \cdot C_{l-1} \mathbb{E}_{(*,j)}^{l-1} \cdot \left( \prod_{k=1}^{l-2} C_k \mathbb{E}_{(*,*)}^k \right) \quad (14)$$

Combining Equation 3 and 14, because the instantiation of the weight matrices and mask matrices for each layer are independent:

$$\mathbb{E}_{[M,W]}(\mathcal{S}_{\text{SF}})^{[l]} = \mathbb{E}\left[ \sum_{i=1}^{C_l} \sum_{j=1}^{C_{l+1}} \mathcal{S}_{\text{SF}}(W_{(i,j)}^l) \right] = \sum_{i=1}^{C_l} \sum_{j=1}^{C_{l+1}} \mathbb{E}\left[ \mathcal{S}_{\text{SF}}(W_{(i,j)}^l) \right]$$

$$= \left( \prod_{k=l+2}^N p_k C_{k+1} \mathbb{E}_{|\mathcal{D}^k|} \right) \sum_{i=1}^{C_l} \sum_{j=1}^{C_{l+1}} \left( p_{l+1} C_{l+2} \mathbb{E}_{(i,*)}^{l+1} \cdot p_l \mathbb{E}_{|\mathcal{D}^l|} \cdot p_{l-1} C_{l-1} \mathbb{E}_{(*,j)}^{l-1} \right) \left( \prod_{k=1}^{l-2} C_k \mathbb{E}_{|\mathcal{D}^k|} \right)$$

$$= \left( \prod_{k=l+2}^N p_k C_{k+1} \mathbb{E}_{|\mathcal{D}^k|} \right) C_l C_{l+1} \left( p_{l+1} C_{l+2} \mathbb{E}_{|\mathcal{D}^{l+1}|} \cdot p_l \mathbb{E}_{|\mathcal{D}^l|} \cdot p_{l-1} C_{l-1} \mathbb{E}_{|\mathcal{D}^{l-1}|} \right) \left( \prod_{k=1}^{l-2} p_k C_k \mathbb{E}_{|\mathcal{D}^k|} \right)$$

$$= C_{N+1} \prod_{l=1}^N (p_l C_l \mathbb{E}_{|\mathcal{D}^l|})$$

$$(15)$$

According to Equation 9,

$$\mathbb{E}_{[M,W]}(\mathcal{S}_{\text{SF}}) = \sum_{l=1}^N C_{N+1} \prod_{l=1}^N (C_l \mathbb{E}_{|\mathcal{D}^l|})$$

$$= N C_{N+1} \prod_{l=1}^N (p_l C_l \cdot \mathbb{E}_{x \sim \mathcal{D}}[|x|]), \quad (16)$$

SynExp Invariance Theorem can also be extended to CNNs, as it is obvious that SynExp of CNNs is proportional to that of fully connected networks. Thus the difference of SynExp between CNNs and fully connected networks for each layer is only a factor equal to $\frac{K^2}{C_{l+1}}$, where $K$ is the kernel size of the convolutional layer. $\qquad\square$

## C ANALYSIS FOR THE ERROR OF SYNEXP PROXY

In the following sections, we first evaluated the SynFlow scores and accuracy of the model pruned by SynFlow as well as other models obtained by randomly shuffling and/or re-initializing the weights of the pruned model. We empirically found that the variance of the accuracy of the models is small, while the variance of the SynFlow scores is large. Then we prove that, under such conditions, as an accuracy proxy, the expectation of the SynFlow Score leads to less error compared to using the SynFlow score.

We first show that the variance of the model accuracy is small if randomly shuffled or reinitialized. The small variance in the accuracy of models has been demonstrated in the previous experiment section. Figures 1, 3, and 5 in the paper depict all experiments on CIFAR-10, with error bars obtained from 3-5 runs of the same experiment. We can clearly see from the figures that there is a small variance in the accuracy of the models pruned by SynFlow.

We further performed the following experiment in the appendix: after pruning with the SynFlow score, we first randomly shuffle the weight mask and randomly initialize the weight, then we evaluate the test accuracy and the SynFlow score of the new model. We repeat the experiment for 10 runs at 4 different sparsities with ResNet-20 on CIFAR-10. We also compute the Spearman correlation coefficient $\rho$ between SynFlow score and accuracy for each run and coefficient of variation $c_v = \frac{\sigma}{\mu}$ for accuracy and SynFlow score.

Table 4: Coefficient of Variation for Accuracy and SynFlow Scores.

| Sparsity | Acc (mean $\pm$ std) | $C_v$ of Acc | SynFlow (mean $\pm$ std) | $C_v$ of SynFlow | $\rho$ |
|---|---|---|---|---|---|
| 0.03 | $76.02 \pm 0.58$ | $7.6 \times 10^{-3}$ | $2.67 \pm 0.36$ | $1.3 \times 10^{-1}$ | 0.16 |
| 0.14 | $85.39 \pm 0.48$ | $5.6 \times 10^{-3}$ | $3.09 \pm 0.42$ | $1.4 \times 10^{-1}$ | -0.19 |
| 0.30 | $88.05 \pm 0.36$ | $4.1 \times 10^{-3}$ | $3.48 \pm 0.45$ | $1.3 \times 10^{-1}$ | -0.08 |
| 0.46 | $88.85 \pm 0.39$ | $4.4 \times 10^{-3}$ | $3.50 \pm 0.52$ | $1.5 \times 10^{-1}$ | -0.54 |

Next, we show that The variance of the SynFlow score is large if randomly shuffled or reinitialized. After randomly shuffling the weight and randomly reinitializing the weight, because the coefficient of variation (CV) of accuracy is much smaller than that of the SynFlow score, we validate that the variance of accuracy is small and the variance of the SynFlow is large.

Remember that Spearman correlation coefficient $\rho \in [-1, 1]$, and it reveals the rank correlation between two random variables. Here, the Spearman correlation coefficient $\rho$ between SynFlow score and accuracy is close to 0, and sometimes it is even negative. It indicates that at the same sparsity, given the higher SynFlow score, we cannot predict model accuracy will be higher or lower; when the Spearman correlation coefficient $\rho$ is negative, a higher SynFlow score is more likely to result in lower model accuracy.

Next, we prove that at a given sparsity level, when the score has a large variance while the accuracy has a small variance, using the expectation of the score leads to a proxy for accuracy with less error compared to using the score as a proxy for accuracy.

**Theorem C.1.** *For any function, $f$ that maps SynFlow score $\mathcal{S}$ to test the accuracy of the model $y$, let $\hat{f}$ be an estimator of $f$, and $h$ be a function represents random perturbation. If random perturbation $h$ does not change the expectation of the accuracy, we have $\mathbb{E}_h[f(h(\mathcal{S}))] = f(\mathcal{S})$. Then we can prove that $\mathbb{E}_h[\hat{f}(h(\mathcal{S})]$ is a better estimation of $f(\mathcal{S})$ compared to $\hat{f}(S)$ in terms of $L_2$ norm error.*

*Proof.*

$$\mathbb{E}_{\mathcal{S}}[\|\mathbb{E}_h[\hat{f}(h(\mathcal{S}))] - f(\mathcal{S})\|^2]$$
$$= \mathbb{E}_{\mathcal{S}}[\|\mathbb{E}_h[\hat{f}(h(\mathcal{S})) - f(h(\mathcal{S}))]\|^2]$$
$$\leq \mathbb{E}_{\mathcal{S}}[\mathbb{E}_h[\|\hat{f}(h(\mathcal{S})) - f(h(\mathcal{S}))\|^2]]$$
$$= \mathbb{E}_{\mathcal{S}}[\|\hat{f}(\mathcal{S}) - f(\mathcal{S})\|^2]$$

Cauchy–Schwarz inequality is used in the second step as $\|\cdot\|^2$ is convex. Equality holds only when $f$ and $\hat{f}$ are linearly independent, which is clearly not the case. $\square$

## D   SOLUTION OF THE OPTIMIZATION PROBLEM

For the convex optimization problem in Equation 5, Equation 6, or PreConfig, we can simply use Karush–Kuhn–Tucker (KKT) conditions to analytically solve it. We include the solutions as follows for completeness. In practice, we use convex solver to solve the problem to avoid the piecewise function.

### D.1   SOLUTION FOR EQUATION 5

$$p_l = \min(\frac{\mu}{\alpha}, 1)$$
$$\text{where } \mu \text{ satisfies: } \sum_{l=1}^{N} \min(\alpha_l, \mu) = B_{\text{params}} \tag{17}$$

### D.2   SOLUTION FOR EQUATION 6

$$p_l = \min(\frac{1}{\mu_1 \alpha_l + \mu_2 \beta_l}, 1)$$
$$\text{where } \mu_1, \mu_2 \text{ satisfy: } \sum_{l=1}^{N} \alpha_l \min(\frac{1}{\mu_1 \alpha_l + \mu_2 \beta_l}, 1) = B_{\text{params}}$$
$$\sum_{l=1}^{N} \beta_l \min(\frac{1}{\mu_1 \alpha_l + \mu_2 \beta_l}, 1) = B_{\text{flops}} \tag{18}$$

### D.3   SOLUTION FOR PRECONFIG

$$p_l = \frac{1}{\mu_1 \alpha_l + \mu_2 \beta_l}$$
$$\text{where } \mu_1, \mu_2 \text{ satisfy: } \sum_{l=1}^{N} \alpha_l \frac{1}{\mu_1 \alpha_l + \mu_2 \beta_l} = B_{\text{params}}$$
$$\sum_{l=1}^{N} \beta_l \frac{1}{\mu_1 \alpha_l + \mu_2 \beta_l} = B_{\text{flops}} \tag{19}$$

In practice, to avoid solving the $\mu$, we use a convex optimization solver, which can obtain the solution with a CPU within a second for such a small scale convex optimization.

## E   MORE EMPIRICAL RESULTS ON SYNEXP INVARIANCE THEOREM

We show more empirical results that validate SynExp Invariance Theorem. We first show the comparison of the performance using different pruning granularities on VGG16 using CIFAR-10. All the settings in this experiment are the same as in Figure 2, except this experiment is done on VGG16.

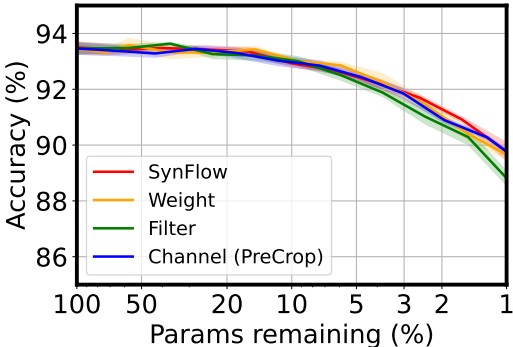

Figure 6: Comparison of the performance using different pruning granularities on VGG16 using CIFAR-10.

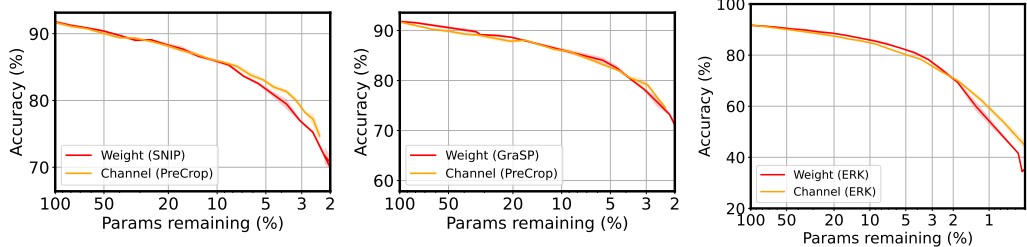

Figure 7: Comparison of the performance using different pruning granularities on ResNet20 using CIFAR-10. SNIP (left), GraSP (middle), and ERK (right) sparsity are used.

Then we also verify that SynExp Invariance Theorem not only holds for SynFlow, but also holds for other PAI algorithms. In this experiment, we first use other PAI (i.e., SNIP and GraSP) to obtain the layerwise density $p_l$. Then we use random pruning to match $p_l$ in the channel level. The results are shown in Figure 7.

As shown in all the above experiments, as long as the layerwise density is the same, the pruning granularties do not affect the model accuracy.

## F  CHANNEL WIDTH COMPARISON

We also include a comparison of the channel width between the baseline EfficientNetB0 and PreConfig EfficientNetB0 in Figure 8.

## G  EXPERIMENT DETAILS

### G.1  IMPLEMENTATION

We adapt model implementations of ResNet, ShuffleNet, and MobileNetv2 from imgclsmob[3]. The implementations of SynFlow, SNIP, and GraSP are based on the codebase of SynFlow[4].

### G.2  HYPERPARAMETERS

Here we provide the hyperparameters used in training all models in Table 5. No AutoAugment, Label Smoothing, or stochastic depth is used during training. All the CIFAR-10 models are trained with same hyperparameter setting.

---

[3]https://github.com/osmr/imgclsmob
[4]https://github.com/ganguli-lab/Synaptic-Flow

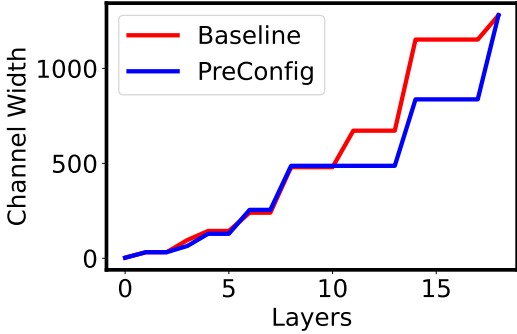

Figure 8: Comparison of the channel width of EfficientNetB0 before and after PreConfig.

Table 5: Hyperparameters used in training.

|  | CIFAR-10 | ImageNet | | |
|---|---|---|---|---|
|  |  | MobileNet | ResNet | EfficientNet |
| Optimizer | momentum | momentum | momentum | momentum |
| Training Epochs | 160 | 180 | 90 | 150 |
| Batch Size | 128 | 256 | 512 | 256 |
| Initial Learning Rate | 0.1 | 0.025 | 0.2 | 0.035 |
| Learning Rate Schedule | linear | drop at each epoch | drop at 30, 60 epoch | drop at each epoch |
| Drop Rate | N.A. | 0.98 | 0.1 | 0.99 |
| Weight Decay | $10^{-4}$ | $4 \times 10^{-5}$ | $10^{-4}$ | $4 \times 10^{-5}$ |

### G.3    COMPUTE COST

The total computational cost incurred in all experiments conducted in this paper amounts to approximately 150 GPU days using the NVIDIA Tesla V100.

### G.4    BENCHMARK SETTINGS

We measure the latency of our pruned model on a 16GB NVIDIA Tesla V100 GPU and a 4-core Arm Cortex A57 CPU. The batch size is set to 128 and 256 on EfficientNet and MobileNetV2, respectively. Both models are running with PyTorch Paszke et al. (2019) v1.7 and cuDNN Chetlur et al. (2014) v10.1.

## H    LIMITATION OF SYNEXP INVARIANCE THEOREM

SynExp Invariance Theorem is applicable exclusively to pruning methods based PAI methods, given our fundamental assumption that weights are randomly initialized, rendering any differences among them negligible. However, there are certain methods to which our theorem does not apply: (1) Training-based pruning methods: These methods identify crucial weights and connections through the process of training, which allows them to make the weights and connections distinguishable. (2) Lottery Ticket Hypothesis (LTH) Frankle & Carbin (2018): as per the experiments highlighted in Frankle et al. (2020), it is observed that the accuracy of the network will be affected, if the network pruned by LTH is shuffled or reinitialized. This proves that, unlike standard PAI methods, LTH is able to identify important weights with the training process at a cost of more expensive training process. Therefore, changing the pruning granularity can have an impact on the accuracy in these settings.

