# OpenReview forum: "Structured Pruning of CNNs at Initialization"
_ICLR.cc/2024/Conference — ICLR 2024 Conference Withdrawn Submission_

### Official Review · Reviewer_gkKF · 2023-10-31

**Soundness:** 2 fair
**Presentation:** 3 good
**Contribution:** 2 fair
**Rating:** 3
**Confidence:** 4

**Summary:**

This paper proposes a method for structured pruning of CNNs at initialization. It introduces "synaptic expectation" as a proxy metric for accuracy. The authors show that irrespective of the granularity of pruning, with the same layer-wise density, the "synaptic expectation" remains the same. Based on this, they propose methods agnostic to the initialized weights and data, directly outputting the layer-wise densities. Specifically, they introduce PreCrop and PreConfig, where the latter can have density constraints larger than 1, leading to the expansion of certain layers.

**Strengths:**

- The manuscript is clearly written and reader-friendly.

- The focus on structured pruning at initialization offers hardware advantages over traditional weight pruning, making this exploration valuable.

- The authors have conducted experiments across a diverse set of models, showcasing broad applicability.

**Weaknesses:**

- The paper uses layer density as the importance score and tries to maximize the sum of the layer densities with the FLOPs/Parameters constraints. This approach raises concerns, as it's an oversimplification to presume homogeneity in densities across layers. For instance, numerous structured pruning techniques for residual networks assign variable layer budgets, often designating lower density (or fewer channels) to the initial layers of residual blocks, aiming to create a bottleneck structure. However, the method described in the paper appears to uniformly maximize layer densities, which might not be optimal.

- The results presented in Figure 5 appear to have an inherent bias. A model can be configured to prune a higher number of parameters while maintaining high FLOPs or the inverse. Thus, it is important for a fair comparison to compare both the FLOPs and parameters of the models at the same time similar to the baselines, rather than two different models optimized separately.

- Pruning fundamentally revolves around removing the less important weights or weight groups. The paper's observations, albeit intriguing, deviate from this principle. The method's independence from weights and data might actually detract from its core purpose. In the absence of these elements, the method essentially evolves into a mere CNN architecture suggester. This is a simplified NAS problem, and although it is much more cost-effective than typical NAS algorithms, the objective of the search (which is to maximize the cumulative log of layer-wise densities) is not strong enough.

**Questions:**

- What is the set of the augmentation techniques used in training the models?
- What are the FLOPs pruned in Figure 5 left and the parameters pruned in the same figure right compared to the other models?

---

### Official Review · Reviewer_mFWq · 2023-11-01

**Soundness:** 2 fair
**Presentation:** 2 fair
**Contribution:** 2 fair
**Rating:** 3
**Confidence:** 5

**Summary:**

This paper formulates Pruning-at-initialization (PAI) methods into a convex optimization problem based on an expectation-based proxy for model accuracy, which can instantly produces the optimal allocation of the layer-wise densities under the proxy model. The author of e paper also extend the method to reducing the DNN model dimension without incurring extra computations.

**Strengths:**

1. PreCrop is a good extend to the current PAI method.

2. The author of the paper talks about the preliminary of PAI in detail, which is very informative and helpful.

**Weaknesses:**

1. The writing of the paper is not clear. Figure 1, as the most important figure in the paper, is not referenced in the main body of the paper. There is no sufficient discussion about Figure 1.

2. The overview of the method (figure 1) is not matching with the description of the PreCrop in the introduction, which is very confusing. For example, what is the random channel pruning in figure 1? The author should talk about it before entering the heavy part of the main body. Otherwise, the paper will confuse readers.

3. Related work does not provide sufficient references. Most citations are before the year of 2020, which is relatively old.

4. The proposed method uses computation-aware constraint training to prune the model, which is not novel.

**Questions:**

Please refer to weaknesses

---

### Official Review · Reviewer_umau · 2023-11-01

**Soundness:** 3 good
**Presentation:** 3 good
**Contribution:** 3 good
**Rating:** 5
**Confidence:** 5

**Summary:**

Model compression is highly useful for deploying the model to the low end device or faster inferences. Pruning is standard an approach approach for the model compression, most of the pruning model requires pretrained model which is costly. The pruning at initialization model does not requires the pretrained weight for the model compression. In this work author proposed PreCorp model which leverages layer-wise-density at initial stage to prune the model (channel). In this work the author proposed layer-wise-density estimation as an optimization problem which can be solved using the convex optimization approach.  The proposed proxy metric called SynExp to address the deficiencies of the existing method seems interesting.

**Strengths:**

[1] The PAI approach is useful, specially in the environment of recent development of the LLM.

[2] The layer-wise density approach seems novel and shows the good results, over the CIFAR-10 and ImageNet dataset over the various architecture.

[3] The model is applicable to filter as well as weight pruning.

[4] The result at the high pruning ration seems interesting.

**Weaknesses:**

[1] The proposed approach seems good but the evaluation is the key weakness, only the Table-1 (ImageNet dataset experiment over the RESNET50, MOBILENETV2, EFFICIENTNETB0) result gives some insite about the proposed approach. As claimed by the author the approach is generic and can be applied to various architecture. I expect to the author the proposed approach should be applied to the transformer architecture like ViT[1] or Swin[2]. In the current scenario transformer architecture is more important and widely applied than the CNN architecture.

[2] The intuition why method works i.e. why layer-wise density is important is not clear? Also, how it will be useful if we want to prune the FC layer or individual weight?

[3] The structured pruning method is important which shows high practical speed up compared to unstructured pruning, here as claimed by the author their approach is structured pruning model, as we know that all the channel pruning is structured, its nothing novel in the channel pruning model.

[4] There are few other approach [3] that pruned the model in the PAI setting and its follows the structured pruning. How the model behave compared to [3]?

[5] Once we computed the layer-wise density the exact pruning step is not clear, also, I believe that reproducibility is hard without the code.

[6] In compared to the model SynFlow, SNIP, Grasp and ERK the proposed model shows similar behavior with a low pruning ration why is differ and has a large gap with high pruning ratio?



[1] An Image is Worth 16x16 Words: Transformers for Image Recognition at Scale, ICLR-2020
[2] Swin Transformer: Hierarchical Vision Transformer using Shifted Windows, ICCV-2021
[3] Pushing the Efficiency Limit Using Structured Sparse Convolutions, WACV-2023

**Questions:**

Please refer to the weakness section.

---

### Official Review · Reviewer_BbJ9 · 2023-11-08

**Soundness:** 3 good
**Presentation:** 3 good
**Contribution:** 3 good
**Rating:** 5
**Confidence:** 3

**Summary:**

This paper proposes a pruning-at-initialization (PAI) method, which (1) extends the SynFlow to a measurement named SynExp that is independent of the pruning granularity and can be used to obtain layer-wise density through convex optimization; (2) a channel-level pruning technique PreCrop is built upon the SynExp; (3) PreCrop is further extended to a method that is capable to increase the number of channels of the original model.

Experiments on CIFAR-10 and ImageNet are conducted.

**Strengths:**

1. Different from previous PAI methods, the proposed method can be used to perform channel-level structural pruning, and the authors theoretically and empirically prove that under the same layer density, the proposed method can obtain model that have similar accuracy compared to weight-level and filter-level sparsity pruning methods. Therefore, the method enjoys significant speedup in real production environment as well as obtaining competitive performance to unstructural pruning methods.

2. The improvements of PreCrop compared to previous SOTA SynFlow are significant on ImageNet.

**Weaknesses:**

1. Given the marginal accuracy improvements and FLOPs reduction on ImageNet in Table 2, PreConfig cannot be treated as an effective contribution of this paper.

2. Though the accuracy improvements compared to SynFlow are significant on ImageNet, PreCrop is still far behind the training-based pruning methods and evaluation-based search methods. For example, [1] achieves no accuracy drop with 50% FLOPs reduction on ResNet-50, while PreCrop has 1.1% accuracy drop with 28% FLOPs reduction. Therefore, my major concern is the practicality of this method.

3. I would like to see experiments on more model variants such as VGG on CIFAR-10.

4. The writing quality needs to be improved. (1) Page 4, no period sign after "Equation 1 into Equation 3"; (2) Some figures do not have titles and values for the x-axis and y-axis; (3) Section D, "solve the problem to avoid the", "to" -> "and".

I gave a initial reject rating due to the concerns in technical contribution, practicality, and writing quality of this paper. I can update my rating if some of my concerns are solved.

[1] Fang, G., Ma, X., Song, M., Mi, M. B., & Wang, X. (2023). Depgraph: Towards any structural pruning. In Proceedings of the IEEE/CVF Conference on Computer Vision and Pattern Recognition (pp. 16091-16101).

**Questions:**

1. In Figure 5, the performance of PreCrop-Params is similar to SynFlow on most of the compression ratios, while PreCrop-FLOPs has obviously better performance when compression ratio is high. Any explanations to it?

2. Can the authors visualize some architectures obtained by PreCrop and PreConfig? I am quite interested on how many layers can have a density level > 1.